# Effective Removal of Tetracycline from Water Using Copper Alginate @ Graphene Oxide with In-Situ Grown MOF-525 Composite: Synthesis, Characterization and Adsorption Mechanisms

**DOI:** 10.3390/nano12172897

**Published:** 2022-08-23

**Authors:** Bing Chen, Yanhui Li, Qiuju Du, Xinxin Pi, Yuqi Wang, Yaohui Sun, Mingzhen Wang, Yang Zhang, Kewei Chen, Jinke Zhu

**Affiliations:** 1College of Mechanical and Electrical Engineering, Qingdao University, 308 Ningxia Road, Qingdao 266071, China; 2State Key Laboratory of Bio-Polysaccharide Fiber Forming and Eco-Textile, Qingdao University, 308 Ningxia Road, Qingdao 266071, China

**Keywords:** MOF-525, graphene oxide, tetracycline, copper alginate, in-situ growth

## Abstract

For nanomaterials, such as GO and MOF-525, aggregation is the main reason limiting their adsorption performance. In this research, Alg-Cu@GO@MOF-525 was successfully synthesized by in-situ growth of MOF-525 on Alg-Cu@GO. By dispersing graphene oxide (GO) with copper alginate (Alg-Cu) with three-dimensional structure, MOF-525 was in-situ grown to reduce aggregation. The measured specific surface area of Alg-Cu@GO@MOF-525 was as high as 807.30 m^2^·g^−1^, which is very favorable for adsorption. The synthesized material has affinity for a variety of pollutants, and its adsorption performance is significantly enhanced. In particular, tetracycline (TC) was selected as the target pollutant to study the adsorption behavior. The strong acid environment inhibited the adsorption, and the removal percentage reached 96.6% when pH was neutral. Temperature promoted the adsorption process, and 318 K adsorption performance was the best under experimental conditions. Meanwhile, 54.6% of TC could be removed in 38 min, and the maximum adsorption capacity reached 533 mg·g^−1^, far higher than that of conventional adsorption materials. Kinetics and isotherms analysis show that the adsorption process accords with Sips model and pseudo-second-order model. Thermodynamic study further shows that the chemisorption is spontaneous and exothermic. In addition, pore-filling, complexation, π-π stack, hydrogen bond and chemisorption are considered to be the causes of adsorption.

## 1. Introduction

Tetracycline (TC) is a broad-spectrum antibiotic, widely used in livestock and humans, that significantly reduces infections and deaths caused by bacteria [1]. However, due to human metabolism, more than 70% of TC is directly discharged into medical wastewater and even domestic wastewater [2]. In addition, TC in aqueous solutions is difficult to remove, and there have been reports of multiple antibiotics detected in drinking water, as well as TC detected in the urine of preschool children [3]. Therefore, it is necessary to investigate the effective control and removal of TC for the sustainable development of human beings. Currently, the removal of TC mainly depends on its physicochemical properties, which commonly include adsorption [4], catalytic degradation [5], advanced oxidation [6], and membrane separation [7]. Among them, adsorption is the most widely used method due to low cost, removal efficiency, and large-scale use [8].

Recently, metal organic frameworks (MOFs), a class of novel crystalline materials, with large specific surface area, adjustable pore structure, and unsaturated metal sites, has shown great application potential in hydrogen storage, nanofiltration, and supercapacitors [9,10]. However, when MOFs are used in water treatment, the interaction between water molecules and unsaturated metal sites leads to the destruction of lattice structure and reduced activity, which limits the application of MOFs [11,12]. Fortunately, previous reports have mentioned that the use of high charge density metal ions can enhance the coordination bond strength, and lattice structures can remain stable in water for a long time, which provides the basis for the application of MOFs materials in water adsorption [13]. PCN-777 [14], UiO-66 [15], and MOF-525 [16], which use Zr^4+^ as the metal ion, have demonstrated excellent water stability. According to the report of Farhad et al. [17], UiO-66 has excellent stability in water, chloroform, and dimethylformamide, and has good adsorption performance for various dyes. Xia et al. [18] applied MOF-525 to adsorb TC, and compared with UiO-66 and NU-1000, MOF-525 was found to have the best pore size and topological structure for adsorbing TC.

Simultaneously, MOFs with poor stability can be combined with nanomaterials to improve their water stability and adsorption performance, which has become a research focus in the field of adsorption [19,20]. Graphene oxide (GO) is preferred because of its high concentration of oxygen-containing functional groups, large specific surface area, and unique carbon structure [21]. In previous studies, the combination of MOF-525 and GO did enhance the adsorption performance of TC [22]. However, MOF-525 and GO tend to cluster, greatly reducing the adsorption performance [23]. Nevertheless, nanomaterials were often coated in the three-dimensional structure of natural polymers to reduce agglomeration [24,25]. But the mass use of polymers also occupies a large number of adsorption sites. In-situ growth method can not only disperse nanomaterials but also reduce the loss in adsorption sites.

Sodium alginate (SA), a promising adsorbent containing a large number of oxygen-containing functional groups, is often used as a fixation material for GO [26]. Due to its water solubility, SA needs to be crosslinked by divalent metal ions before adsorption. Due to the complexation of Cu^2+^ and TC, copper alginate (Alg-Cu) has the best adsorption performance for TC [27,28]. In addition, GO can also load Cu^2+^, and the removal performance is also improved [29]. On this basis, in order to obtain adsorbents that can remove a large amount of TC, MOF-525 will be grown in-situ on Alg-Cu@GO, which not only reduces the aggregation of GO and MOF-525 but also reduces the occupation of MOF-525 adsorption active sites. The synthesis process diagram of Alg-Cu@GO@MOF-525 is shown in Figure 1.

The successful synthesis of Alg-Cu@GO@MOF-525 was determined by SEM, XRD, BET, FTIR, and TGA, and its physical and chemical properties were characterized. Although the specific surface area of Alg-Cu@GO obtained after grinding was only 1.6 m^2^·g^−1^, the specific surface area of the adsorbent (MOF-525 grown in-situ) was up to 807 m^2^·g^−1^, providing favorable conditions for the adsorption of pollutants. In addition, adsorption experiments for different dyes were also carried out in the study, which proved that Alg-Cu@GO@MOF-525 has good performance for various pollutants. In order to study the adsorption process and performance of Alg-Cu@GO@MOF-525 for TC in detail, the adsorption experiment was designed by different variables and adsorption kinetics, adsorption isotherms, and adsorption thermodynamics were studied. Finally, the adsorption mechanism was also discussed.

## 2. Materials and Methods

### 2.1. Materials

Sodium alginate (99.9%, SA) was purchased from Shanghai Aibi Chemical Reagent Co., Ltd. (Shanghai, China). Copper sulfate anhydrous (CuSO_4_, 99%) was purchased from Tianjin Regent Chemicals Co., Ltd. (Tianjin, China). Acetone (C_3_H_6_O, 99.9%), benzoic acid (C_7_H_6_O_2_, 99.5%), zirconium(IV) oxychloride octahydrate (ZrOCl_2_·8H_2_O, 99%), and N, N-dimethylformamide (C_3_H_7_NO, 98%, DMF) were purchased from Sinopharm Chemical Reagent Co., Ltd. (Shanghai, China). Additionally, tetracids(4-carboxyphenyl) porphyrin (97%, H_4_TCPP) was purchased from Shanghai bidepharmatech Co., Ltd. (Shanghai, China). Meanwhile, deionized water was used as the only water source in the whole experiment. All other chemical reagents were purchased commercially and used directly without further purification.

### 2.2. Synthesis

GO was prepared by the modified Hummers process as we reported earlier and used after vacuum drying [30]. Synthesis of Alg-Cu@GO: Firstly, 3.0 g of SA was dissolved in 100 mL deionized water and stirred vigorously for 12 h. Next, 1.8 g (0 g, 0.6 g, 1.2 g) of GO were added, assisted by ultrasound, and stirred for 24 h to obtain a uniform mixture. Subsequently, the mixture was injected into a 0.3 M CuSO_4_ solution using a syringe, fully crosslinked for 12 h and then washed with deionized water to obtain Alg-Cu@GO hydrogel beads. Finally, the hydrogel beads were dried in an oven at 333 K for 12 h and fully ground. Synthesis of Alg-Cu@GO@MOF-525: This method is an improvement on the hydrothermal synthesis of MOF-525, enabling MOF-525 to grow in-situ on Alg-Cu@GO [31]. In brief, 2.0 g of benzoic acid and 0.105 g of ZrOCl_2_·8H_2_O were added to a 20 mL vial with 8 mL DMF. Next, the mixture was dissolved uniformly by ultrasound and heated at 353 K for 2 h. After cooling naturally to room temperature, 0.053 g of H_4_TCPP and 0.15 g of Alg-Cu@GO (with different GO content) were added, and the additive was dispersed uniformly by ultrasound. Next, the mixture was hydrothermal synthesized in an oven at 353 K for 24 h. The purplish red suspension was obtained after the reaction completed, and Alg-Cu@GO@MOF-525 was obtained after centrifugation. Subsequently, repeated washing with DMF and acetone were used to remove excess impurities. Finally, after vacuum drying for 2 h and activation at 393 K, it can be used for the adsorption experiment.

### 2.3. Characterization

The surfaces morphology of the adsorbents was characterized by scanning electron microscopy (SEM, Supra55, Zeiss, Oberkochen, Bartenburg, Germany), and the element distribution was characterized by extended components (EDS). The crystal structures were obtained by X-ray diffractometer (D8 Advance, Bruker, Karlsruhe, Baden-Wurttemberg, Germany) with Cu-Kα source (λ = 1.5406 Å). The scanning range was between the values of 5° and 30°. After 12 h of dehydration and desorption at 353 K and vacuum, the N_2_ adsorption–desorption isotherms were tested at 77 K using an automatic specific surface area analyzer (ASAP 2460-2M, Micromeritics, Norcross, GA, USA), and the BET equation was used to calculate the specific surface area of the sample. After deducting the spectral lines with air background, a FTIR spectrometer (IS50, Thermo Fisher Scientific, Waltham, MA, USA) was used to characterize the presence of functional groups. Thermal decomposition and stability of samples were measured at a heating rate of 10 K·min^−1^ in the 303-973 K range using a thermogravimetric analyzer (Discovery SDT 650, TA, New Castle, DE, USA).

### 2.4. Adsorption Experiments

All the solutions involved in the adsorption experiments were obtained by diluting the standard stock solution of 1000 mg·L^−1^. Meanwhile, the adsorption experiment was placed in a water bath shaker with a preset temperature (298 K) and rotation speed (200 rpm), and the whole process was kept dark to reduce the degradation of pollutants. The adsorption properties were evaluated after 48 h. Then, the residual concentration was measured using a UV-Vis spectrophotometer (TU-1810, Beijing General Instruments Co., Ltd., Beijing, China) after centrifugation. The calculation formula of adsorption capacity (mg·g^−1^) at equilibrium (*q_e_*) and predetermined time (*q_t_*) is as follows:(1)$$q_{e}=\frac{\left(c_{0}-c_{e}\right)}{m}\times V$$
(2)$$q_{t}=\frac{\left(c_{0}-c_{t}\right)}{m}\times V$$
where *c*_0_, *c_e_*, and *c_t_* represent the initial concentration before adsorption, the final concentration when adsorption reaches equilibrium, and the solution concentration measured at the pre-set time (mg·L^−1^), respectively; *V* signifies the volume of the solution used (L); and *m* is the dosage of the adsorbent added (g).

Adsorption experiments were carried out by setting variables, including dosage, pH, temperature, initial concentration, contact time, and different pollutants. Generally, the amount of adsorbent was 0.5 mg·mL^−1^, and the initial concentration was 200 mg·L^−1^. Under the conditions, the adsorption properties of all synthesized adsorbents for TC were compared. The range of dosage study was 2-20 mg (2, 4, 7, 10, 15, and 20 mg), and the range of pH study was 2-8. When temperatures as the variable, 298 K, 308 K, and 318 K are selected, and the experiment was completed along with different initial concentration, which ranged from 80-280 mg·L^−1^ with a gradient of 40 mg·L^−1^. In the investigation with contact time as variable, 200 mL TC solution was used. Methylene blue, acid fuchsin, methyl orange, and congo red have also been used as pollutants to study the adsorption properties of Alg-Cu@GO@MOF-525. The molecular sizes were obtained after structural optimization using the Forcite module of molecular simulation software Materials Studio 2019 (Accelrys, Qingdao, China).

## 3. Results and Discussion

### 3.1. Characterization

Figure 2a displays the microstructure of MOF-525, and its regular hexahedral structure is one of the important characteristics to judge its existence [32]. Moreover, the microstructure of Alg-Cu is given in Figure 2b. There are no obvious pores on the surface of Alg-Cu, instead, a large number of fold structures are generated due to intense contraction, which is consistent with the results previously reported [33]. In addition, the uniform distribution of C, O, and Cu indicates that Cu^2+^ is fully crosslinked (Appendix A). Figure 2c and Appendix A illustrate the microscopic surface morphology of Alg-Cu@GO. Compared with the dense surface of Alg-Cu, the addition of GO makes the surface smoother and presents a wrinkled structure and fracture, which may be due to the GO intercalation in the structure of Alg-Cu and uneven shrinkage in the drying process leading to the occurrence of fracture. In elemental distribution, the increase in Cu content may be due to the large amount of adsorption of Cu^2+^ by Alg-Cu@GO at high concentration [34]. In Figure 2d,e, it is apparent that cubic crystals densely cover the surface of the substrate made from Alg-Cu@GO. These crystal structures are very similar to MOF-525 in Figure 2a, supporting the successful synthesis of Alg-Cu@GO@MOF-525. However, the particle size and distribution of MOF-525 grown in-situ on it are disordered, which may be caused by the uncontrollable particle size and surface of Alg-Cu@GO. The uniform distribution of Zr, O, and C in the EDS distribution also proves that the crystal structure is MOF-525. Due to the large amount of MOF-525 coverage, thus Cu content is greatly reduced.

The crystalline structures of Alg-Cu@GO@MOF-525 was analyzed by XRD and the obtained XRD patterns are presented in Figure 3a. The XRD patterns of Alg-Cu and Alg-Cu@GO showed no obvious diffraction peaks, which may be the loss in lattice structure caused by dehydration. The characteristic peak of GO at 2θ = 11.3° disappears due to ultrasound (Appendix A) [35]. Alg-Cu@GO@MOF-525 and MOF-525 have similar spectra, and the characteristic peaks belonging to MOF-525 were measured at 2θ = 6.4°, 7.9°, 9.1°, 11.1°, and 13.9°, corresponding to plane (011), (111), (002), and (112), respectively. The characteristic peaks measured by MOF-525 were consistent with those previously reported, which further confirms the success of MOF-525 in-situ growth on Alg-Cu@GO [36].

The specific surface area of Alg-Cu, Alg-Cu@GO, and Alg-Cu@GO@MOF-525 were calculated using N_2_ adsorption–desorption isotherms. The specific surface area of Alg-Cu and Alg-Cu@GO was calculated to be 1.52 and 1.60 m^2^·g^−1^, respectively, and no pore structure was detected. Thus, in-situ growth of MOF-525 may occur only on the outer surface of Alg-Cu@GO, and the load-bearing surface can only be increased by fine grinding. Nevertheless, the specific surface area of the synthesized Alg-Cu@GO@MOF-525 jumped to 807.30 m^2^·g^−1^. This is attributed to the huge specific surface area of MOF-525, which makes the BET surface area of the in-situ grown material increase dramatically. The adsorption curve was consistent with the type-I isotherm, and the adsorption capacity increased rapidly at low pressure, which was the result of the micropores of the adsorbent being completely filled with condensed liquid. As can be seen from the pore size distribution in the Figure 3b, the pores in Alg-Cu@GO@MOF-525 were highly developed, the pore volume was up to 0.43 cm^3^·g^−1^, and the pores were mainly smaller than 2 nm (1.18 nm and 1.58 nm), which further proved that the synthesized materials were microporous materials.

FTIR spectra provide a basis for the investigation of functional groups in adsorbents. The spectral lines of Alg-Cu and Alg-Cu@GO were almost the same, and there were three characteristic peaks at 1590, 1440, and 1032 cm^−1^, respectively (Appendix A), which belong to the asymmetric and symmetric stretching vibration of -COO^−^ and the stretching vibration of C–O, respectively [37]. The disappearance of the peak of GO at 1735 cm^−1^ may be due to the hydrogen bond interaction with Alg-Cu, as well as the interaction with Cu^2+^ [34]. Meanwhile, MOF-525 and Alg-Cu@GO@MOF-525 have similar FTIR characteristic peaks, and the peaks at 1603 and 1413 cm^−1^ are attributed to the in- and out-plane stretching vibration peaks caused by carboxylate groups [38]. The characteristic peak near 1556 cm^−1^ may be caused by the stretching vibration of C=C of benzene ring in TCPP, and the peak at 1603 cm^−1^ may also be the result of benzene ring vibration. Meanwhile, the characteristic peaks related to Zr–O bond and the characteristic peak related to C–N bond were also found at 1177, 719, and 662 cm^−1^ and 1347 cm^−1^, respectively [18]. Additionally, the bending vibration peak of -OH and stretching vibration peak of -C–O were found at 1150 cm^−1^ and 1000 cm^−1^, respectively. However, the difference is that the stretching vibration peak of C=O appears in Alg-Cu@GO@MOF-525 at 1715 cm^−1^, which may be the reason for the combination of Alg-Cu@GO and MOF-525. After the adsorption of TC, FTIR spectrum analysis was performed on the material again. The results showed that the carbonyl peak disappeared, which may be related to the binding mode of TC and adsorbent, and the peak changes in other peaks indicated that TC were successfully adsorbed on Alg-Cu@GO@MOF-525.

The thermogravimetric curves from 30 °C to 850 °C were given by Figure 3d. From the initial temperature to 150 °C, Alg-Cu@GO loses 8% of its weight. The evaporation of physically and chemically bound water molecules is the main reason for this weight loss. The heat flow curve also indicates that this stage is dehydration. When the temperature is higher than 150 °C, the heat flow curve shows the melted state with decomposition, and the weight loss in this part is 35%, which should be the result of the fracture of 1,4-glycosidic bond [39]. In the subsequent process, the thermal weight loss rate of the material gradually flattens, which may be the result of the gradual decomposition and melting of the organic components in the material, and the final mass residual is 32%. Compared to Alg-Cu, Alg-Cu@Go@MOF-525 is more thermally stable, with water completely removed at 165 °C (and only 2.5%), and the heat flow peak of decomposition melting does not appear until 312 °C. In Alg-Cu@GO@MOF-525, due to the decrease in Alg-Cu ratio, the thermal weight loss in 1,4-glycosidic bond is only 22.5%, and the weight loss near 535 °C may be caused by the oxidative decomposition of organic ligand skeleton. Finally, due to the in-situ growth of MOF-525, the final residual weight of the composite is 46%.

### 3.2. Bath Experiment

#### 3.2.1. Effect of GO Content on Adsorption Performance

Figure 4a shows the experimental results of the effects of different GO contents on the adsorption properties. The increase in GO content in Alg-Cu@GO has a slight improvement on the adsorption performance, which may be because the benzene ring structure in GO can form π-π stacking with the benzene ring structure in TC, increasing the adsorption performance. However, when the addition ratio is low, it may be because GO is completely coated and the effective adsorption sites are less exposed, thus reducing the adsorption performance. Simultaneously, the adsorption performance of Alg-Cu@GO@MOF-525 was greatly improved by 3.4, 5.0, 3.7, and 3.3 times compared with Alg-Cu@GO, respectively. The results revealed that the increase in GO content did improve the adsorption properties, possibly due to the increased exposure of adsorption sites and enhanced the stability and adsorption driving force of MOF-525. In addition, 60% of GO content was used for the subsequent experiments and characterization in this study.

#### 3.2.2. Effects of pH, Dosage, Temperature, and Contact Time on Adsorption Properties

The pH value is an important factor affecting the adsorption performance and has great influence on the physical and chemical properties of adsorbents and adsorbates. TC is easy to deactivate under alkaline conditions, thus the experiment was carried out under acidic and weakly alkaline conditions. Moreover, TC has positive, neutral, and negative charges at different pH values [40]. Figure 4b shows the adsorption performance of Alg-Cu@GO@MOF-525 for TC at different pH values. When pH value is less than 3.3, TC mainly exists in the form of TCH_3_^+^, which not only forms competitive adsorption with H^+^ but also has electrostatic repulsion due to protonation of oxygen-containing functional groups [41]. Experimental results also show that the adsorption performance is poor under such conditions. When the pH value is greater than 3.3, the adsorption performance is greatly improved. When the pH value is 7, the adsorption performance is the best, and the removal rate is up to 96.6%. In this pH range, TC is neutral, the electrostatic repulsion between TC and adsorbent decreases, and the competitive adsorption also gradually decreases. When pH is greater than 7, TC begins to have a negative charge, and the deprotonated adsorbent will generate electrostatic repulsion with TC, thus the adsorption performance decreases. Therefore, the electrostatic interaction in the process of TC onto Alg-Cu@GO@MOF-525 mainly exists in the form of electrostatic repulsion.

The amount of adsorbent used is another important factor, and the results will provide a basis for practical use. There are also fewer available adsorption sites at low dosage, resulting in a low removal rate. However, due to the high residual concentration, TC can be more easily adsorbed on the surface of the adsorbent, and the equilibrium adsorption capacity reached 700 mg·g^−1^. With the increase in adsorbent dosage, the number of adsorption sites also increased, which promoted the improvement of removal rate but resulted in a significant decrease in adsorption capacity. When the amount of adsorbent is greater than 15 mg, TC is almost removed and the additional adsorption sites are no longer used for adsorption, so the removal rate does not change [42].

Figure 4d shows the effect on initial concentration of TC solution and temperature on adsorption performance. With the increase in initial concentration, the adsorption capacity increases gradually, which may be because the unoccupied adsorption sites are gradually used. Meanwhile, with the higher the residual concentration, TC is more inclined to be adsorbed on the adsorbent. Temperature is also an important factor. It can be seen from the results that the adsorption capacity increases gradually with the increase in temperature, which indicates that the adsorption process may be an endothermic process.

Figure 4f depicts the change in adsorption capacity with contact time. Although the contact time does not affect the adsorption capacity at equilibrium, it is necessary to explore the control steps and investigate the adsorption mechanism. In the initial stage of adsorption, 54.6% of TC was removed within 38 min. At this stage, a large number of adsorption sites are not occupied, so TC is easily adsorbed onto Alg-Cu@GO@MOF-525. With the increase in time, the adsorption sites on the surface are gradually occupied, and TC molecules need to further diffuse into the adsorbent. Therefore, the adsorption rate gradually slows down, and the removal rate needs 15 h to remain unchanged.

#### 3.2.3. Alg-Cu@GO@MOF-525 Adsorption Properties of Different Dyes

Dye wastewater is also an important source of environmental pollution. Figure 4f shows the results of adsorption of AF, MO, MB, and CR. Compared with Alg-Cu and Alg-Cu@GO, Alg-Cu@GO@MOF-525 has the best adsorption performance for different types of dyes. Although it does not have selective adsorption, its wide adsorption performance also has great application prospects. In addition, the dye molecules are not only different in structure, but charge properties are also different, AF and CR belong to anionic dyes, and MO and MB belong to cationic dyes. The addition of GO improves the adsorption capacity of Alg-Cu@GO for cationic dyes but decreases the adsorption capacity of anionic dyes, which is consistent with the adsorption capacity of GO itself in previous studies [43]. Compared with GO, MOF-525 can improve the adsorption properties of various dyes, independent of the charge they carry. This is similar to the results of the study on the influence of pH on adsorption performance, where electrostatic interaction is not the main reason affecting the adsorption of pollutants onto Alg-Cu@GO@MOF-525.

### 3.3. Adsorption Kinetics

Adsorption kinetics is an important part of adsorption mechanism and can be used to investigate the substance transfer and rate control. In order to accurately explain the adsorption process, four adsorption kinetic models (pseudo-first-order model, pseudo-second-order model, Elovich model, and intra-particle diffusion model) were used to compare the fitting degree and confirm the optimal model. Meanwhile, all parameters and correlation coefficients in the process of adsorption kinetics fitting are listed in Table 1, and the fitting curves are shown in Figure 5.

Pseudo-first-order model describes the process in which pollutants are adsorbed to the surface by overcoming surface resistance through physical diffusion and physical interaction [44]. Pseudo-second-order model considers that chemical interaction is the main factor controlling the adsorption rate, and the adsorption process dominated by chemical adsorption generally has the highest fitting degree [45]. Elovich model is suitable for the adsorption process with irregular adsorbent surface and higher reaction activation energy [46]. The nonlinear fitting equation of pseudo-first-order model, pseudo-second-order model, and Elovich model are given in Equations (3)–(5) as follows:(3)$$q_{t}=q_{e}\left(1-e^{-kt}\right)$$
(4)$$q_{t}=\frac{q_{e}v_{0}t}{q_{e}+v_{0}t}$$
(5)$$q_{t}=\frac{1}{\beta }\ln\left(\alpha \beta t\right)$$
where *k* refers the constant related to the adsorption rate, *t* is the preset measurement time corresponding to *q_t_*; *v_0_* is related to the initial adsorption rate when adsorbent is added; and *α* and *β* are the constants related the adsorption rate.

The fitting results show that the correlation fitting coefficients of pseudo-first-order model, pseudo-second-order model, and Elovich model are 0.940, 0.992, and 0.975, respectively. Therefore, pseudo-second-order model is the best model to explain the adsorption process, and the fitting results are consistent with the experimental data. The adsorption process of TC on Alg-Cu@GO@MOF-525 can be considered as chemisorption.

Moreover, the adsorption process can be divided into several stages: firstly, the pollutant diffuses around the adsorbent; then, overcoming the resistance of the liquid, diffuses to the surface of the adsorbent; next, diffuses from the outer surface to the inner surface; and finally, binds to the adsorption site [47]. Since the initial diffusion process and the binding to the adsorption site are independent of the properties of the adsorbent, membrane diffusion and internal diffusion are the main factors controlling the adsorption process. The division of the adsorption process can be obtained by multi-segment linear fitting curve, which is the intra-particle diffusion model. Equation (6) is the formula of the diffusion model in particles, as follows:(6)$$q_{t}=k_{i}t^{\frac{1}{2}}+C_{i}\;$$

The adsorption process was divided into three parts by linear fitting. The slope of the first stage is larger, which should correspond to the film diffusion, where TC rapidly diffuses to the surface of the adsorbent. The adsorption rate of the second and third stages gradually slows down, and internal diffusion begins to replace membrane diffusion as the main control mechanism of the adsorption process. The second phase is likely to be diffused by TC into the pore structure of MOF-525, and the third phase is likely to be diffused by TC into the substrate of Alg-Cu@GO. Since the intercepts of the three stages do not pass through the origin, thus the adsorption rate is controlled by internal diffusion and membrane diffusion [48].

### 3.4. Adsorption Isotherms

In order to further reveal the interaction mechanism between Alg-Cu@GO@MOF-525 and TC, the experimental data were nonlinear fitted by adsorption isotherm model, and the maximum adsorption capacity was obtained under experimental conditions. The parameters and correlation coefficients involved in the calculation used are presented in Table 2, and the fitting curves are shown in Figure 6.

The Langmuir model assumes that the adsorbent is ideal; that is, the active sites are uniformly distributed and have the same adsorption capacity, and the pollutants are independent of each other [24]. The Freundlich model is an empirical model derived from experimental results and used to describe multilayer adsorption processes with irregular adsorbent surfaces [49]. The Sips model is derived from the Langmuir and the Freundlich models, and the introduced *n_s_* value make the Sips model describe the adsorption process more extensively [50]. When *n_s_* = 1, the Sips model is the Langmuir model, which describes homogeneous monolayer adsorption. When *n_s_* < 1, multilayer adsorption on irregular surfaces can be described. The nonlinear equations used are as follows:(7)$$q_{e}=\frac{q_{m}bc_{e}}{1+bc_{e}}$$
(8)$$q_{e}=K_{f}c_{e}^{1/n}$$
(9)$$q_{e}=\frac{q_{m}{\left(b_{s}c_{e}\right)}^{n_{s}}}{1+{\left(b_{s}c_{e}\right)}^{n_{s}}}$$
where *b*, *K_f_*, and *n* are the constant related to the adsorption affinity of adsorbent; *b_s_* is an improved constant related to the adsorption site, and *n_s_* reflects the heterogeneity of the adsorbent.

The fitting curve showed that the adsorption data fitted well with the Langmuir model, and the correlation coefficients were 0.977, 0.983, and 0.992, respectively. Thus, the surface of Alg-Cu@GO@MOF-525 may be uniform and TC was monolayer adsorbed on the surface. Meanwhile, the Freundlich model also has a high fitting correlation coefficient but lower than the Langmuir model. However, n value is an important indicator reflecting whether it has the adsorption intensity. The n values obtained at the three temperatures are all between 1 and 10, which proves that Alg-Cu@GO@MOF-525 is easy to adsorb TC. Compared with the above models, the Sips model has the highest correlation fitting coefficient and is higher than the Langmuir model. The Sips model is considered to be the best model to explain the adsorption process. The results show that the calculated ns values are all less than 1, which indicates that the adsorption of TC onto Alg-Cu@GO@MOF-525 surface is multilayer, and the adsorbent is not uniform. In addition, the calculated maximum adsorption capacity at 318 K is as high as 533.2 mg·g−1, which is significantly improved compared with the commonly used adsorbent. Table 3 lists the maximum adsorbents reported for TC adsorption. The maximum adsorption capacity of Alg-Cu@GO@MOF-525 is up to five times higher than that of magnetic GO and nearly two times higher than that of Fe3O4@MOF-525. It is also improved compared with MOF-525 and MOF-525/GO under approximate conditions. Meanwhile, it is suggested that in-situ grown MOF-525 on substrates, which are not easy to aggregate, is a new strategy to improve the adsorption performance.

### 3.5. Thermodynamic Study

The adsorption of TC onto Alg-Cu@GO@MOF-525 can be further studied by using Gibbs free energy change and calculating the thermodynamic parameters of the adsorption process. The formulas used for calculation are as follows:(10)ΔG=ΔH−TΔS
(11)$$\ln\left(\frac{q_{e}}{c_{e}}\right)=-\frac{\Delta H}{RT}+\frac{\Delta S}{R}$$
where R is the universal gas constant (8.314 J·mol^−1^·K^−1^). *ΔG*, *ΔH*, and *ΔS* stand for Gibbs free energy change, enthalpy change, and entropy change, respectively. The values of Δ*H* and Δ*S* can be received directly from the linear fitting of ln(*q_e_*/*c_e_*) to 1/*T*. In addition, Table 4 shows the thermodynamic parameters at different temperatures.

Δ*G* is less than zero at all temperatures, indicating that the adsorption process can occur spontaneously. Moreover, the absolute value of Δ*G* increases with the increase in temperature and the positive value of Δ*H* (31.6 kJ·mol^−1^) indicates that the TC adsorb onto Alg-Cu@GO@MOF-525 occurs more easily with higher temperature, and the adsorption process is endothermic. The positive value of Δ*S* reflects the random increase in the solution interface after adsorption [55].

### 3.6. Possible Adsorption Mechanism

In previous studies, MOF-525 was considered to be suitable for TC adsorption due to its suitable topological structure and pore matching well with TC molecule [18]. MOF-525 grown in-situ on Alg-Cu@GO enables the synthesized material to also have the advantages of large specific surface area and suitable pores. Therefore, the pore-filling between TC molecule and in-situ MOF-525 is one of the main reasons for the greatly improved adsorption performance [56]. This also explains the differences in the adsorption of different dye molecules. The molecular lengths of MB, TC, MO, and CR may differ, but other sizes are similar, which may be the reason why they can be adsorbed in large numbers. However, due to the large size of AF molecule, it could not enter the cage structure of MOF-525, and the adsorption performance was poor. (See Appendix A)

From the influence of pH on the adsorption performance, the charge carried by the adsorbent is similar to that carried by TC in the process of adsorption, resulting in electrostatic repulsion, which may be another reason affecting the adsorption. This effect is weak because dye molecules with different charges do not cause huge changes in adsorption properties.

Both Zr^4+^ in MOF-525 and Cu^2+^ in Alg-Cu@GO can bind TC through complexation, which makes Alg-Cu@Go@MOF-525 adsorb a large amount of TC in a short time [27,57]. Meanwhile, the benzene ring structure in the organic ligand can form a strong interaction of π-π stack with the benzene ring in TC molecule, which may also be the reason for the good adsorption performance of Alg-Cu@GO@MOF-525 [26]. In addition, the abundance of -OH and -COOH groups on Alg-Cu and GO provide active sites for hydrogen bonding between TC and adsorbents [58].

In the above adsorption model fitting, the adsorption process is more consistent with the Sips model and the pseudo-second-order model, and it is considered that the adsorption is a multi-layer adsorption dominated by chemisorption. Meanwhile, large enthalpy changes in thermodynamic studies also indicate the presence of chemisorption, and this chemisorption is endothermic. By comparing the changes in functional groups before and after adsorption by FTIR, it was speculated that the disappearance of C=O might be caused by chemisorption. Finally, the Zr-O cluster with structural defects in MOF-525 can also have acid–alkaline interaction with the amine group in TC [59].

## 4. Conclusions

In this work, Alg-Cu@GO@MOF-525 was obtained by in-situ growth of MOF-525 on the basis of Alg-Cu@GO, which reduced the performance degradation caused by MOF-525 and GO aggregation. As a result, the specific surface area was as high as 807.30 m^2^·g^−1^, and the pore distribution ranged from 10 Å to 20 Å, which was very favorable for the adsorption of small-size pollutants. The adsorption results of various pollutants show that the synthesized Alg-Cu@GO@MOF-525 has good adsorption performance for various pollutants. Among them, a detailed study has been carried out on TC, and the material with 60% of GO has the best adsorption performance and the best adsorption performance at pH = 7. Meanwhile, 54.6% of TC was removed within 38 min, and the maximum adsorption capacity was 533 mg·g^−1^. In addition, the model fitting shows that the adsorption process of TC onto Alg-Cu@GO@MOF-525 was in accordance with the Sips model and the pseudo-second-order model. The adsorption process is considered to be multilayer surface adsorption with chemical adsorption as the main process, and thermodynamic study shows that the adsorption process is spontaneous endothermic process. Pore-filling, complexation, π-π superposition, hydrogen bond and chemical adsorption are the main reasons of adsorption. Electrostatic repulsion is also involved in the adsorption process.

## Figures and Tables

**Figure 1 nanomaterials-12-02897-f001:**
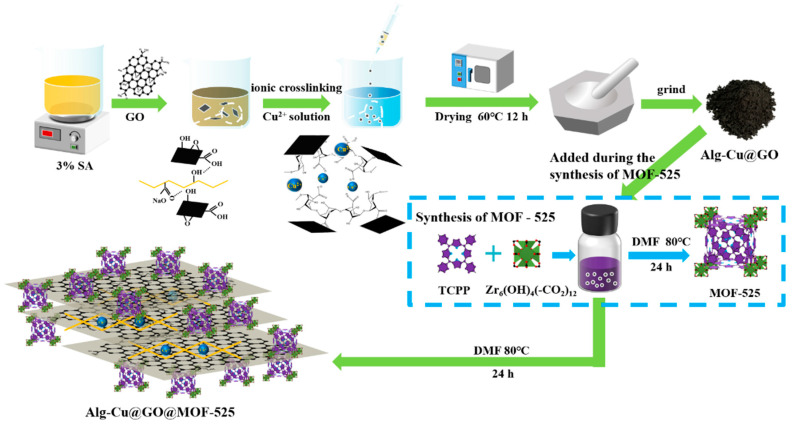
Schematic diagram of synthesis process of Alg-Cu@GO@MOF-525. (SA: Sodium alginate; GO: Graphene oxide; TCPP: Tetracids porphyrin; DMF: N, N-dimethylformamide).

**Figure 2 nanomaterials-12-02897-f002:**
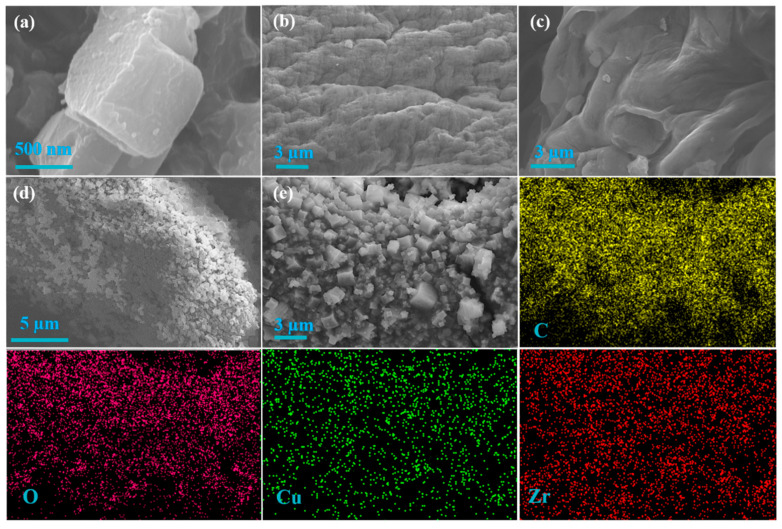
SEM images of MOF-525 (**a**), Alg-Cu (**b**), and Alg-Cu@GO (**c**); SEM images of Alg-Cu@GO@MOF-525 and the elemental distributions of C, O, Cu, and Zr (**d**,**e**).

**Figure 3 nanomaterials-12-02897-f003:**
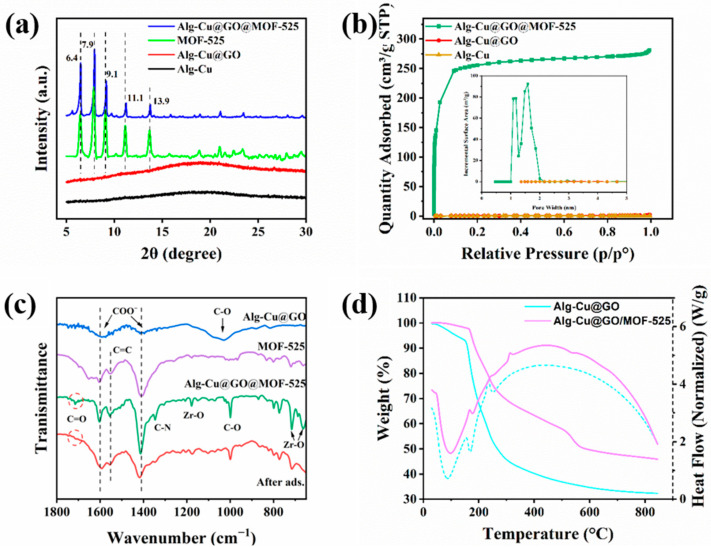
(**a**) XRD patterns, (**b**) N_2_ adsorption–desorption isotherms, (**c**) FTIR spectra, and (**d**) TGA curves of the synthesized adsorbents.

**Figure 4 nanomaterials-12-02897-f004:**
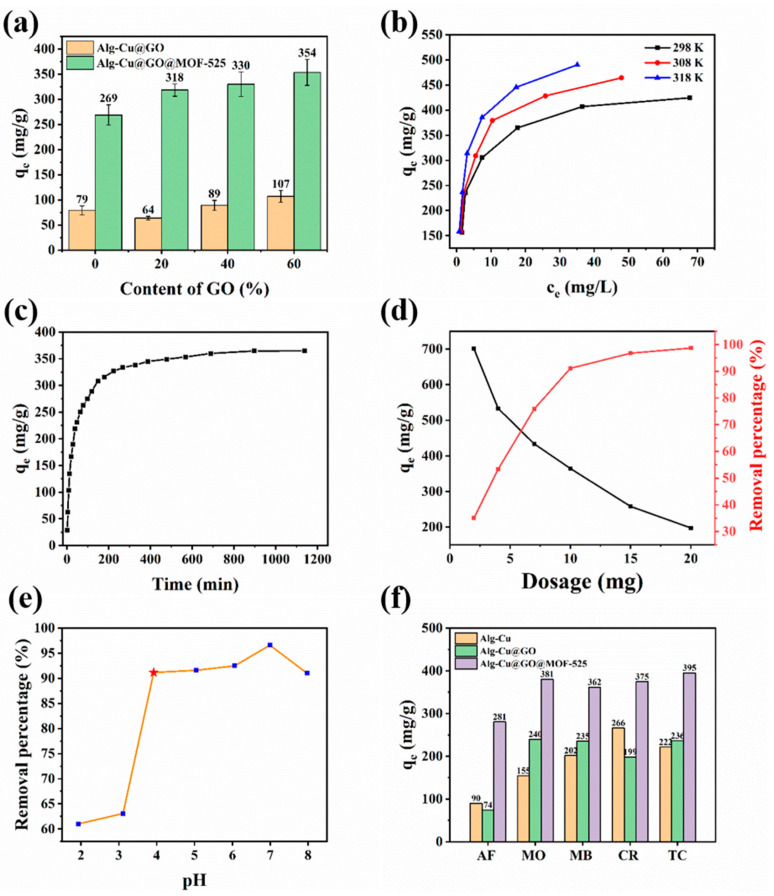
Effect of different experimental parameters on the adsorption of TC by Alg-Cu@GO@MOF-525: (**a**) GO content, (**b**) pH, (**c**) dosage, (**d**) temperature, (**e**) contact time, and (**f**) different dye molecules (AF, MO, MB, CR).

**Figure 5 nanomaterials-12-02897-f005:**
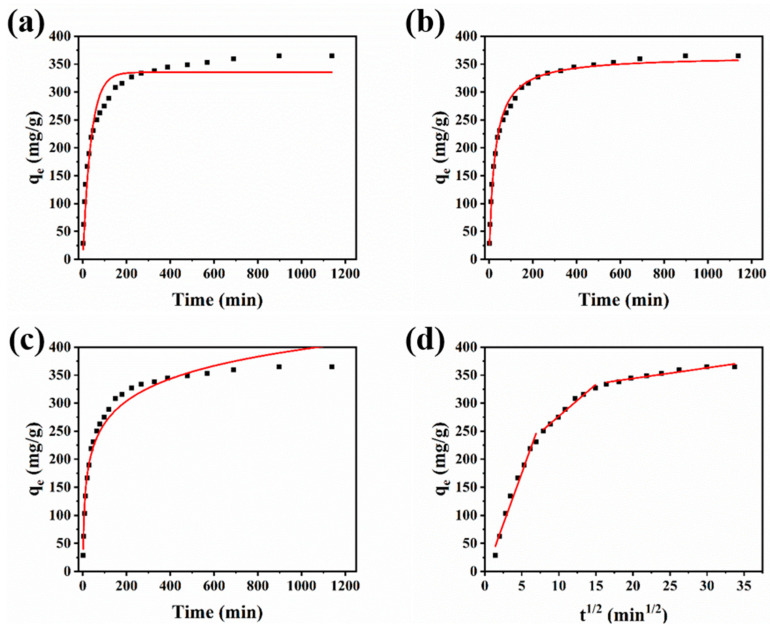
Adsorption kinetics of TC adsorbed by Alg-Cu@GO@MOF-525: (**a**) pseudo-first-order model, (**b**) pseudo-second-order model, (**c**) Elovich model, and (**d**) intra-particle diffusion model.

**Figure 6 nanomaterials-12-02897-f006:**
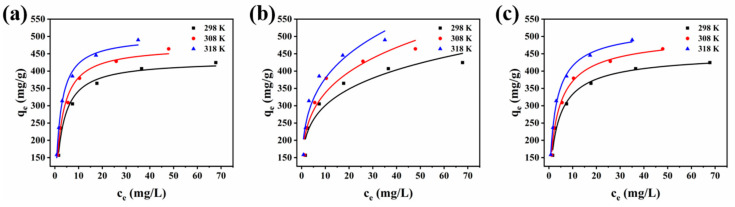
Adsorption isotherms of TC adsorbed by Alg-Cu@GO@MOF-525: (**a**) Langmuir model, (**b**) Freundlich model, and (**c**) Sips model.

**Table 1 nanomaterials-12-02897-t001:** Fitting results of kinetic model parameters for TC onto Alg-Cu@GO@MOF-525.

Kinetic Model	Parameters	Values
Pseudo-first-order model	*k* (min^−1^)	0.03
	*q_e_* (mg·g^−1^)	335.4
	R^2^	0.940
Pseudo-second-order model	*v_0_* (mg·g^−1^·min^−1^)	14.31
	*q_e_* (mg·g^−1^)	364.8
	R^2^	0.992
Elovich model	*α*	57.38
	*β*	0.020
	R^2^	0.975
Intra-particle diffusion model	*k_1_* (mg·g^−1^·min^−1/2^)	36.47
	*C_1_*	−6.685
	R^2^	0.973
	*k_2_* (mg·g^−1^·min^−1/2^)	11.37
	*C_2_*	162.9
	R^2^	0.974
	*k_3_* (mg·g^−1^·min^−1/2^)	1.896
	*C_3_*	306.0
	R^2^	0.913

**Table 2 nanomaterials-12-02897-t002:** Parameters and correlation coefficients for the adsorption of TC on MOF-525/GO.

Models	Parameters	298 K	308 K	318 K
Langmuir isotherm	*q_m_* (mg·g^−1^)	430.7	472.7	501.1
	*b* (L·mg^−1^)	0.39	0.42	0.53
	R^2^	0.977	0.983	0.992
Freundlich isotherm	*K_f_* (mg^1−1/n^ L^1/n^·g^−1^)	195.6	215.7	184.7
	*n*	4.23	4.08	4.72
	R^2^	0.919	0.923	0.901
Sips isotherm	*q_m_* (mg·g^−1^)	455.9	511.6	533.2
	*b_s_* (mg^−1/n^·L^−1/n^)	0.340	0.340	0.460
	*n_s_*	0.810	0.790	0.830
	R^2^	0.977	0.988	0.996

**Table 3 nanomaterials-12-02897-t003:** Maximum adsorption capacity of different adsorbents.

Adsorbents	Dosage (mg·mL^−1^)	*c_0_* (mg·mL^−1^)	*q_m_* (mg·g^−1^)	*T* (K)	Ref.
MGO ^a^	0.067	5–50	106.6	313	[51]
UiO-66-(COOH)_2_/GO	0.5	10–100	165.0	298	[52]
Fe_3_O_4_@MOF-525	0.1	10–200	277.0	328	[53]
Alg/Fe_3_O_4_@C@TD ^b^	1.7	20–100	476.2	298	[54]
MOF-525	0.5	100–200	372	303	[22]
MOF-525/GO	0.5	100–200	436	303	[22]
Alg-Cu@GO@MOF-525	0.5	80–240	533.2	318	This work

^a^ magnetic graphene oxide; ^b^ encapsulation of Fe_3_O_4_@maltose-functionalized triazine dendrimer in alginate.

**Table 4 nanomaterials-12-02897-t004:** Thermodynamic parameters for TC adsorbed by Alg-Cu@GO@MOF-525.

*T* (K)	Δ*G* (kJ·mol^−1^)	Δ*H* (kJ·mol^−1^)	Δ*S* (J·mol^−1^ K^−1^)
298	−4.6	31.6	121.5
308	−5.8	-	-
318	−7.0	-	-

## Data Availability

The data presented in this study are available on request from the corresponding author.

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
