# Peer review of "Effective Removal of Tetracycline from Water Using Copper Alginate @ Graphene Oxide with In-Situ Grown MOF-525 Composite: Synthesis, Characterization and Adsorption Mechanisms"

_nanomaterials, 2022, doi:10.3390/nano12172897_

Round 1
Reviewer 1 Report
In this paper, Chen and coworkers studied Alg-Cu@GO@MOF-525 composite for Tetracycline(TC) adsorption in the aspect of different variables, adsorption kinetics, thermodynamics, and adsorption mechanism. As the authors mentioned, water purification is an environmentally important issue for the human being. To remove the TC in the water, the authors suggested a composite design of highly porous material with a large surface area. Therefore, they used MOF material with graphene oxide. However, the TC adsorption of MOF-525 and Alg-Cu used in this study was proved in the previous report. There is no novel in the material itself. Nonetheless, their systematic studies of adsorption performance, kinetics, and thermodynamics are valuable. Therefore, I recommend accepting this manuscript in Nanomaterials. In addition, I have a small comment that the authors need to address.
In the adsorption experiment, we can clearly confirm the role of MOF-525 in all data. However, the authors did not include pure MOF-525 as the control experiment. I recommend that the author add the performance of pure MOF-525. If all high-performance composite is comparable to pure MOF-525, it is an unnecessary design approach. The authors said Zr4+ of MF-525 and Cu2+ in Alg-Cu can bind TC. Is there any direct evidence? Isn’t there any effect of porphyrin organic linker in MOF-525?
Reviewer 2 Report
The authors present a new composite material (Alg-Cu@GO@MOF-525) for the removal of tetracycline from water. The study of the adsorption capability under various conditions is comprehensive and the mechanism presented appears reasonable. The improvements over the state-of-the-art need to be more thoroughly treated (details below, point 6).
I would recommend publication only after the following revisions:
1. The English is quite good, but please check again because some sentences are grammatically incorrect (e.g. the first sentence of the introduction, lines 32-34), and some vocabulary is unclear (specifically, “scene requirements”, line 43)
2. Line 19 in abstract: expand TC to tetracycline (TC)
3. Fig. 1 is unclear after the grinding step. Is the blue arrow what happens without Alg-Cu@GO and the green arrow what happens with it or are they sequential, i.e. DMF 80 °C 24h followed by the same treatment again (this seems unlikely from the description in the methods section)? Please clarify in the figure
4. In Fig. 2:
a. the EDS spectra are not legible, please re-draw them
b. why is the C signal (map and spectrum) lower rather than higher for Alg-Cu@GO compared to Alg-Cu without GO? Please comment
c. do you have any images that show evidence of the GO sheets within the structure, perhaps at higher magnification? This could support your hypothesis about the different adsorption performance at different GO concentrations due to the exposure (or not) of adsorption sites (discussion in lines 259-270). GO sheets are usually visible in composites like this: if they are not here, please explain why. In fact, in the whole paper the presence of GO is only confirmed indirectly by different adsorption capabilities, because it is not observed directly by SEM, XRD, or FTIR. Some direct confirmation would be useful
5. In Fig. 4 the notation is not consistent: 4a uses Cu-SA@GO instead of Alg-Cu@GO as in the rest of the paper. Please correct
6. In Table 3, there are no entries for MOF-525 alone and GO/MOF-525 composite, which could be taken from references [28] and [32] respectively. In lines 417-418, you state that “The maximum adsorption capacity of Alg-Cu@GO@MOF-525 is…nearly two times higher than that of MOF-525”. Are you referring to the Fe3O4@MOF-525 included in the table? This is not a true value for MOF-525 alone. In ref [28], the maximum adsorption capacity of MOF-525 is in fact a lot higher, 807 mg g-1. If this is true, please explain (with experimental evidence), the real advantages of your Alg-Cu@GO@MOF-525 system. If you do not agree with that value, please explain why, repeating your adsorption experiments on MOF-525 alone as a control.
Reviewer 3 Report
Chen et al. have synthesized, characterized and evaluated the adsorption mechanism of copper alginate@graphene oxide@MOF-525 for effective removal of tetracycline and some dyes. By and large this work has been carried out methodically with all the necessary adsorption parameters to elucidate the adsorption mechanism. However, the following points need to be addressed before this paper could be accepted for publication in Nanomaterials.
1. “in-situ grown of MOF-525” should be “in-situ grown MOF-525”.
2. TC is not defined at the first instance in the abstract.
3. “Tetracycline” is missing the keywords. Also, is it “graphite oxide” or “graphene oxide”.
4. Figure 1 – provide the full form of various abbreviations either directly in the schematic labeling or in the Figure 1 caption.
5. L96, L99, L101, L129, L148 – for all the chemicals, reagents and instruments/equipment mention in section 2, please ensure to provide state, city and country name in the case of USA as well as city and country name in the case of other countries in the indicated sentences and other locations.
6. Avoid small one or two sentence dividing into paragraphs. Instead several small paragraphs could be combined to few large paragraphs.
7. L143 – “reserve” should be “stock”.
8. L146 – “what’s more” should be “Then,”
9. L148-149 – “..adsorption capacity at adsorption equilibrium (qe, mg.g-1) and at predetermined time (qt, mg.g-1) is as follows” should be corrected as “..adsorption capacity (mg.g-1) at equilibrium (qe) and predetermined time (qt) is as follows”.
10. L160 – “…15, 20)” should be “…15 and 20 mg)”.
11. Section 2.4 – the name, version, city and country of purchase of software used for modeling should be provided.
12. Figures 3-6 – the axis and legend labels should be increased in font size for clarity and readability.
13. Figure 3 – the key peak assignment values should be included directly shown in the each of this figure itself.
14. Figure 4f – enough space should be provided between each dye, that is, grouping of three bars should be separated by enough space.
15. L277 - “PH” should be “pH”.
16. L336-341 – these model mechanism implications should be supported by the appropriate reference citation.
17. L349 – “about” should be “related”.
18. L389 – “isothermal” should be “isotherm”.
19. L407-409 – indicate that “n” is “adsorption intensity”.
20. Figure 6b – why the nonlinear Freundlich modeling curve not extended down to the first plotted point?
21. Table 3 – the adsorbents should be explained in full form in the footnote of this table.
22. Line 453 – “electrostatic repulsion may be” should be “electrostatic repulsion which may be”.
23. Line 463 – “In the above model fitting investigate” should be “In the above adsorption model fitting”.
24. References – At least 10 references should be removed from the total number of references.
Round 2
Reviewer 2 Report
Comments addressed sufficiently